

# Dynamic motion monitoring of a 3.6 km long steel rod in a borehole during cold-water injection with distributed fiber-optic sensing

Martin P. Lipus[1], Felix Schölderle[2], Thomas Reinsch[3,1], Christopher Wollin[1], Charlotte M. Krawczyk[1,4], Daniela Pfrang[2], Kai Zosseder[2]

[1] GFZ German Research Centre for Geosciences, Telegrafenberg, 14473 Potsdam, Germany
[2] Technical University Munich, Hydrogeology and Geothermal Energy, Arcisstr. 21, 80333 Munich, Germany
[3] present address: Fraunhofer IEG, Fraunhofer Research Institution for Energy Infrastructures and Geothermal Systems IEG, Am Hochschulcampus 1 IEG, 44801 Bochum, Germany
[4] Technical University (TU) Berlin, Institute for Applied Geosciences, Ernst-Reuter-Platz 1, 10587 Berlin, Germany

*Correspondence to*: Martin P. Lipus (mlipus@gfz-potsdam.de)

**Abstract**

Fiber-optic distributed acoustic sensing (DAS) data finds many applications in wellbore monitoring such as e.g. flow monitoring, formation evaluation, and well integrity studies. For horizontal or highly deviated wells, wellbore fiber-optic installations can be conducted by mounting the sensing cable to a rigid structure (casing/tubing) which allows for a controlled

landing of the cable. We analyze a cold-water injection phase in a geothermal well with a 3.6 km long fiber-optic installation mounted to a ¾" sucker-rod by using both DAS and distributed temperature sensing (DTS) data. During cold-water injection, we observe distinct vibrational events (shock waves) which originate in the reservoir interval and migrate up- and downwards. We use temperature differences from the DTS data to determine the theoretical thermal contraction and integrated DAS data to estimate the actual deformation of the rod construction. The results suggest that the rod experiences thermal stresses along

the installation length – partly in the compressional and partly in the extensional regime. We find strong evidence that the observed vibrational events originate from the release of the thermal stresses when the friction of the rod against the borehole wall is overcome. Within this study, we show the influence of temperature changes on the acquisition of distributed acoustic/strain sensing data along a fiber-optic cable suspended along a rigid but freely hanging rod. We show that observed vibrational events do not necessarily originate from induced seismicity in the reservoir, but instead, can originate from stick-

slip behavior of the rod construction that holds the measurement equipment.

## 1. Introduction

Fiber-optic distributed sensing in bore-hole applications has gained a lot of attention in the recent years. Distributed temperature sensing (DTS) has been used to assess rock thermal properties and locations of water-bearing fractures (e.g.

*Hurtig, 1994*, *Förster, 1997*). DTS was used to perform cement job evaluations and wellbore integrity analysis during and after production tests (e.g. *Pearce et al., 2009*, *Bücker and Großwig, 2017*). The performance of a borehole heat exchanger was monitored with DTS to evaluate the heat input along the wellbore and to measure the regeneration time after a heat



extraction period (*Storch et al., 2010*). While DTS has found its way as a standard tool for wellbore monitoring over the last two decades, the utilization of distributed acoustic sensing (DAS) is still subject to many research questions. *Johannessen et al., 2012* introduced the potential and capabilities for acoustic in-well monitoring applications based on DAS systems which range from e.g. flow measurements, sand detection, gas breakthrough, leak detection to vertical seismic profiling (VSP). *Daley et al., 2013*, *Mateeva et al., 2014*, *Harris et al., 2016*, *Daley et al., 2016* and *Henninges et al., 2021* compare traditional geophone with DAS recordings acquired during a vertical seismic profiling campaign (VSP). *Götz et al., 2018* report on a multi-well VSP campaign at a carbon dioxide storage site by using only one single DAS interrogator. *Finfer et al., 2014* performed an experiment to study DAS applications for turbulent single-phase water flow monitoring in a steel pipe. *Bruno et al., 2018* investigate the potential to use downhole DAS data for cross-hole monitoring between two adjacent wells by inducing low frequency pressure pulses to detect high conductivity zones by measuring characteristic vertical strain patterns. *Naldrett et al., 2018* compare fiber-optic technology to traditional production logging tools and provides field data examples of flow monitoring based on both DTS and DAS with wireline-type installations. *Ghahfarokhi et al., 2019* analyze an extensive data set including borehole geophone and DAS during hydraulic fracturing (cable behind casing) to study micro-seismicity and low frequency events in the borehole. *Raab et al., 2019* shows that DAS data from a behind casing installation can be correlated to conventional cement-bond-long (CBL) recordings by analyzing the acoustic data in noisy drilling and testing operations. *Chang et al., 2020* and *Martuganova et al., 2021* report on reverberating signals in DAS recordings which can occur on free-hanging cables in geothermal wells during fluid injection and which are probably caused by bad cable-to-well coupling. In all reported cases, the coupling of the sensing glass fiber to the surrounding media plays a crucial role for the application of DAS technology.

Especially for the monitoring of deformations occurring over longer time periods, i.e. from minutes to hours to days, the coupling of cable and surrounding environment becomes essential to derive any meaningful result from fiber-optic strain sensing. Where as *Reinsch et al., 2017* provide a theoretical approach to describe the response of the sensing fiber in dependence of the specific cable design, the coupling of the cable to the rock formation strongly depends on the specifics of a measuring experiment. *Lipus et al., 2018* compare data from fiber-optic strain sensing and data from conventional gamma-gamma-density wire-line log during a gravel packing operation in a shallow well for heat storage. *Sun et al., 2020* demonstrate with a laboratory and field test that the extent of a deformed reservoir sandstone and silt caprock by injected $CO_2$ can be quantitively evaluated using static distributed strain sensing over periods of 42 hours (cable behind casing). *Zhang et al., 2020* provide an attempt to use distributed strain sensing to monitor elastic rock deformation during borehole aquifer testing to derive hydraulic parameter information. *Miller et at., 2018* compare DTS and time-integrated DAS recordings from a borehole and finds a correlation between DTS recordings and very low frequent DAS strain recordings. In their work, they report on repeating "slip events" seen in the DAS data as short and confined vibrational events upon temperature changes in the well.

The study at hand observes similar "slip events" and shows their causal connection to the thermo-mechanical response of the borehole construction to water flow therein.



Installing a fiber-optic cable in a borehole requires specialized equipment. Depending on the aim of the fiber-optic monitoring campaign, different cable installation types are possible. One way is to permanently install the cable by mounting it to the outside of a casing and run it together with the casing into the well and cement it in place (e.g. *Henninges et al. 2005*, *Reinsch*
*et al., 2013*, *Lipus et al., 2021*). A cemented fiber-optic cable generally provides a thorough mechanical coupling to the surrounding structure which is favorable for DAS data quality. Due to its placement behind the casing, the fibers do not interfere with well operations and monitoring of the well can be performed at any time. However, the cemented annulus of a well is a crucial secondary barrier element for well integrity which is compromised by the installation of a fiber-optic cable. A fluid pathway could potentially be created along the cable.  cases where the well completion design includes liner elements, a
permanent cable installation behind casing to the end of the well is technically not possible, or at least, very challenging. In such cases, other installation types are available. A semi-permanent installation along e.g. a production tubing or a temporary installation via a wireline cable or coiled tubing allow cable placements inside the borehole after drilling is finished. *Munn et al., 2017* present a field test of a novel "flexible borehole coupling technique" that allows deploying fiber-optic cables in boreholes after completion has finished with an improved mechanical coupling compared to lose installed fiber-optic cables.
Due to physical constrains, this technology is best suited for shallow boreholes (< 425 m). *Becker et al. 2017* provide an analysis of borehole fracture displacements using such kind of cable coupling technique. Another method to land a fiber-optic cable into a well is by mounting it to a rigid rod (e.g. a pump sucker-rod). The stiff sucker-rod acts as a centralizer and guides the flexible fiber preventing it from coiling up. Such type of installation is especially advantageous when the cable should be placed in a deep and deviated well.


To utilize acquired fiber-optic data from a free-hanging/free-lying rod with the highest possible confidence, it is important to understand the behavior of such a long and stiff structure inside a well. Heating and/or cooling of the well will lead to thermal stresses in the material which potentially result in contraction or expansion of the sucker-rod and fiber-optic cable construction. As the fiber-optic cable is firmly attached to the rods, these dynamics influence the distributed strain and temperature sensing.
From DTS monitoring, *Schölderle et al., 2021* found that measurement equipment in the previously described setting does indeed contract upon the injection of cold water and that the points spatially sampled by the distributed sensing change their position. Besides a detailed analysis based on DAS and DTS data of the rod's dynamics in response to temperature changes during a cold-water injection, we show that the resulting thermal stresses are released by the observed vibrational events thus indicating stick-slip like behavior of the rod-borehole wall compound.


## 1.1  Well description and cable installation

The fiber-optic cable is installed within a production well at the geothermal site Schäftlarnstraße in Munich, Germany. A detailed description of the geothermal site and the cable installation procedure is presented in *Schölderle et al., 2021*. The well
was completed with a 20" anchor casing, a 13 3/8", a 9 5/8" liner and a perforated 7" production liner. An overview of the



landing depths is presented in Table 1. The design of the borehole completion is schematically shown in Figure 3 (right subplot). The well is vertical to a depth of 250 m. Below 250 m, the well is slightly inclined to 4° down to a depth of 879 m TVD (880 m MD). A number of kick-off-points (KOP) are located along the well path. These are also listed in Table 2. In the result section, a survey shows the well path. From a flow-meter log it is known, that the most prominent feed zone in the well

is just below the transition from 9 5/8" liner to 7" perforated liner in the depth interval between 2825 - 2835 m MD.

**Table 1: Well design at geothermal site Schäftlarnstraße, Munich (see also Figure 3)**

| Drill bit Ø | Type | Casing/liner Ø | Top (TVD / MD) [m] | Bottom (TVD / MD) [m] |
|---|---|---|---|---|
| | Stand-pipe | 30" | surface | 59.1 / 59.1 |
| 26" | Anchor casing | 20" | surface | 866.2 / 867.5 |
| 17 ½" | Liner | 13 3/8" | 766.0 / 767.0 | 1812.3 / 2010.0 |
| 12 ¼" | Liner | 9 5/8" | 1740.0 / 1907.2 | 2408.7 / 2819.0 |
| 8 ½" | Perforated liner | 7" | 2412.2 / 2810.1 | 2932.7 / 3716.0 |

| | KOP | Inclination [°] | Depth (TVD / MD) [m] | Direction [°] |
|---|---|---|---|---|
| | #1 | 44 | 879 / 880 | 287 |
| | #2 | 42 | 1819 / 2220 | 250 |
| | #3 | 58 | 2432 / 2850 | 250 |
| | #4 | 57 | 2775 / 3432 | 231 |

The downhole fiber-optic cable is a tubing-encapsulated-fiber (TEF) that contains two multi-mode and two single-mode fibers.

In this fiber-in-metal-tube (FIMT) construction, the sensing fibers are embedded in gel and placed in a metal tube. At elevated strain levels, the gel deforms plastically and allows for a relative motion between fiber and cable. Also, creep between cable construction and optical fibers can occur. Strain measurements with such a type of cable are typically applicable for dynamic strain changes (high frequencies) and low deformations (*Reinsch et.al, 2017*). For longer periods and higher deformations, fiber-optic strain sensing with FIMT cables is still possible but it becomes less localized due to deformation of the material. A

laboratory experiment on the relative motion between cable structure and optical fiber in a FIMT cable at higher mechanical stress over time is presented in literature (*Lipus et al., 2018*). The cable has a total nominal diameter of 0.43 inch (1.1 cm) and the cable mantle is made of polypropylene. The cable was landed in the well after drilling was finished. To safely and effectively navigate the placement of the fiber-optic cable down to the end of the almost 3.6 km long well, the cable was strapped to steel rods (sucker rods) which were installed in the well together with the cable. The steel sucker rod also helps to

retrieve the cable from the bore-hole when needed. Due to the high deviations in the well at depth, the cable needs to be gently pushed into the well. Therefore, the rigid sucker rod is used for the installation instead of a wireline-type installation. The final landing depth of the sucker rod construction is 3691 m (MD). Figure 1 depicts the configuration of the sucker rod/fiber-optic



compound. Together with a number of cross-over elements and the final landing joint, more than 400 of individual sucker rod elements were installed in the well. In the following, we refer to the sucker rod / fiber-optic cable construction as "the rod".

The depth reference for the DTS (spot warming) and DAS (tap test) are set to surface.

A fiber-optic pressure/temperature (p/T) gauge was installed with the rod and positioned at the top of the reservoir section at 2755 m (MD).

## 1.2 Monitoring campaign


The data shown in this study was measured before and during a cold-water injection test in a geothermal well. Before the start of fluid injection, the well was shut-in for 29 days, so that the initial temperature profile is close to the natural geothermal gradient of the Bavarian Molasse basin (see *Schölderle et al., 2021*). The temperature at the well head was 17 °C and increasing up to 110 °C at the bottom of the well just before the injection start (see profile "00:48" in left panel in Figure 4). Cold-water

fluid injection started on January 23, 2020 at 00:56 by pumping water through the wellhead which leads to a cooling of the well. With an initial water table at a depth of 170m below surface, water was injected from the surface without pressure built-up at the wellhead. The cold-water injection was maintained for 24h at a flow rate of 83 m3/h. In this study, we analyze the transient phase of well temperature change for the first 72 minutes of cold-water fluid injection.

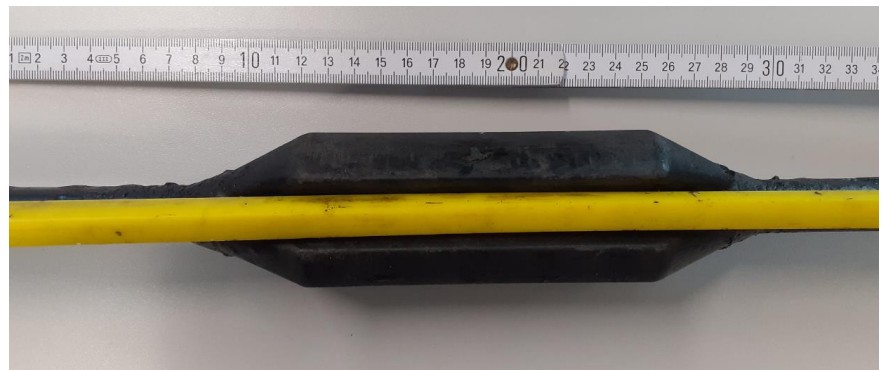


**Figure 1: Down-hole cable configuration of the sucker rod with a centralizer (black) and the fiber-optic cable (yellow)**



## 2. Data Analysis

The analysis in this study is based on the comparison of strain derived from fiber-optic distributed temperature sensing (DTS) and distributed acoustic sensing (DAS).

### 2.1 Derivation of strain from Distributed Temperature Sensing

DTS uses each location of a glass fiber as a sensor for temperature (*Hartog, 1983, Hartog and Gamble, 1991*). This is achieved by coupling laser-light pulses into a glass fiber and analyzing the Raman spectrum of the backscattered light whose origin along the fiber is determined by the two-way travel time of the light. In this study, we use a system based on Raman backscatter. Temperature profiles were acquired every 10 minutes with a spatial sampling of 0.25 m. Detailed information about the performance of the fiber-optic system and the calibration procedure are presented in *Schölderle et al., 2021*.


We calculate the change in temperature from DTS at the start of fluid injection and the profile later during fluid injection. From the temperature change $\Delta T$, a theoretical thermal contraction of the rod is calculated by multiplying $\Delta T$ with the thermal expansion coefficient $\alpha_{rod}$ of the rod. We compare this theoretical thermal contraction with strain information inferred from DAS measurements along the rod. We then use the DTS data to compute stresses along the rod which occur due to cooling.
We compare the result with a static friction curve that deduced from the sucker rod tally and borehole inclination. Using this approach, we can find an explanation for sudden vibrational events which occur irregularly along the rod during the cold-water injection. These events are shown and characterized accordingly.

### 2.2 Direct measurement of strain via DAS


Similar to DTS, DAS also analyzes the back scatter of light coupled into a fiber from one end. Upon contraction or dilatation, the strain-rate of the fiber, i.e. the temporal derivative of relative change of length, can be derived from the temporal change of the interference pattern of coherent light elastically scattered (Rayleigh scattering) from adjacent points within a certain interval of fiber called the gauge length (*Masoudi et al., 2013*). The centroid of the gauge length is defined as a sensor node.
The location (x) of a sensor node along the fiber is again determined by the two-way travel time of light from its source to the node and back. In our study, DAS data is acquired at 1000 Hz, at a gauge length of 10 m, and a spatial sampling of 1 m.





**2.3 Deformation balance from DTS and DAS measurements**


From DTS measurements we may predict themo-mechanical deformation according to

$$\varepsilon_{DTS}(x) = \alpha_{rod} \cdot \Delta T(x) \quad (1)$$

where $\alpha_{rod}$ is the thermal expansion coefficient and $\Delta T(x)$ is the temperature difference at two subsequent points in time at some location x of the fiber. The rod construction as a whole consists of many different materials with different thermal expansion coefficients, such as the sensing fibers, gel filling, metal tubes, polypropylene mantle, steel rod and nylon centralizers. However, the steel of the sucker rod and the steel of the fiber optic mantle are the dominant material by weight and the most relevant for any thermal stresses. The sucker rod consists of 4332 SRX Nickel Chromium Molybdenum steel

with a thermal expansion coefficient of 10 - 13 µε/K (*Hidnert, 1931*) and a modulus of elasticity of 200 GPa (*T.E. Toolbox, 2012*). The second most dominant material is the polypropylene cable mantle with a modulus of elasticity of 1.5-2 GPa (*T.E. Toolbox, 2012*). The proportion of steel on the thermal stresses in the rod construction are 99.8%. For simplicity, we assume that thermal expansion coefficient $\alpha_{rod}$ = 13 µε/K for the sucker rod / fiber-optic cable construction and neglect the other materials.

In contrast to DTS, DAS directly yields the temporal derivative of strain. In order to convert the measured strain rate $\dot{\varepsilon}(x,t)$ data to strain $\varepsilon_{DAS}(x)$ at each location, we integrate in time:

$$\varepsilon_{DAS}(x) = \int_{t1}^{t2} \dot{\varepsilon}(x,t)dt \quad (2)$$

where t1 and t2 delineate the time window and $\dot{\varepsilon}(x,t)$ the recorded strain rate at position x. In the following we speak of

"measured strain" $\varepsilon_{DAS}$ in contrast to "predicted or expected" strain $\varepsilon_{DTS.}$

**2.4 Stick-slip approach**

As the thermal contraction of the cooled sucker rod inflicts a sliding movement of the rods along the borehole wall, we must

consider the friction of their relative motion. This friction would yield a stick-slip motion which is observed almost everywhere when two solid objects are moving relative to one another. A detailed review of the origins of stick-slip behavior in mechanical parts as well as an experimental and theoretical analysis on stick-slip characteristics is presented in literature (e.g. *Berman et al., 1995*). In the simplest case, a stick-slip motion appears when the static friction force $F_f$ between two stationary solid bodies is overcome. A schematic drawing of the forces on an interval of the sucker-rod construction at a depth with borehole

inclination is presented in Figure 2.





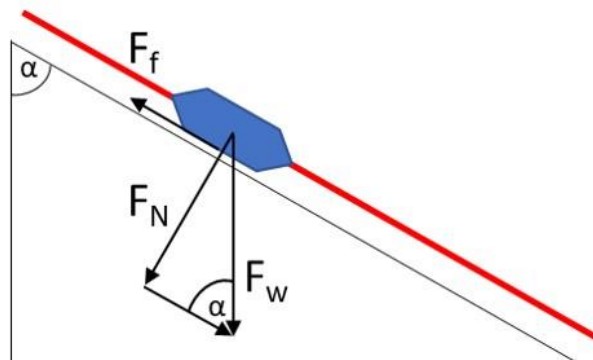

**Figure 2: Static friction force F$_f$ and normal force F$_N$ applying on a sucker-rod contact point (nylon centralizer) as a function of the weight force F$_w$ and the borehole inclination 90°-α**


The static friction force F$_f$ can be calculated according to

$$F_f = \mu \cdot F_N \qquad (3)$$

where F$_n$ is the normal force and μ the static friction coefficient. The value for μ = 0.36 was obtained from a plate-to-plate

experimental analysis on the stick-slip behavior between steel and glass fiber-reinforced nylon specimen (*Muraki et al., 2003*). The force F$_N$ is calculated according to

$$F_N = F_w \cdot sin\alpha = g \cdot m \cdot sin\alpha \qquad (4)$$

where F$_w$ is the gravitational weight force and α the borehole inclination. Each sucker rod element is 9.1 m long, weights 15.7 kg and is equipped with four nylon centralizers and the fiber optic cable (20 g/m). Therefore, the weight force for each contact point of the rod construction yields F$_w$ = 9.81 m/s$^2$ * 15.9 kg / 4 = 39.0 N. Regarding the lowermost part of the rod construction as an example, this means that for the last nylon centralizer (borehole inclination of 54°), a static friction force of F$_f$ = 0.36 · 39.0 N · sin (54°) = 11.3 N is calculated. With respect to contraction of an initially unstressed rod construction, for each

subsequent nylon centralizer towards the surface, the friction force of the rod at the given depth is calculated by the cumulative sum of all friction forces from the nylon centralizers below. The friction force increases with decreasing well depth. Two further weights are added to the friction force profile: the bottom end of the sucker rod is a 1.4 m long steel piece with a weight of 64 kg and the carrier of the pT gauge at 2755 m MD is a 2.2 m long steel piece with a weight of 105 kg. Here, we applied a static friction coefficient for steel on steel of μ = 0.8 (*Lee and Polycarpou, 2007*).






The expected thermal contraction $\varepsilon_{DTS}$ can also be translated to a force. Assuming a Young's modulus for stainless steel of $E$ = 200 GPa (*Materials Handbook, 2008*), we can calculate the stress $\sigma$:

$$\sigma = \frac{F_{app}}{A_{rod}} = E \cdot \varepsilon = E \cdot \varepsilon_{DTS} \qquad (5)$$


given the cross-sectional area of the rod ($A_{rod}$ =2.9 cm$^2$), we can calculate the applied force $F_{app}$ at each location along the rod which was thermally induced within the investigated one-hour cold-water injection period. For simplicity, we assume that the elasticity from the fiber-optic cable and the nylon centralizers are neglectable and that the steel dominates the mechanical behavior of the structure. Furthermore, we make the assumption that no mechanical stresses are exerted on the rod prior to the cold-water injection. This allows us to set a zero-force baseline before injection start for the stick-slip analysis.


**2.5 Event detection and picking**

In the DAS data we monitored repeating vibrational events with ongoing cold-water injection in the deeper part of the well. These events are characterized by a sudden DAS amplitude peak at some depth and an up- and downward directed move-out. With time, the spatio-temporal distribution of these vibrational events changes. To automate the detection of depth location and moveout of an event, we employ a short-term/long-term average trigger (*Allen, 1978, Vaezi and v.d. Baan (2015)*). The parameters used for the STA/LTA analysis can be found in Table 2:



**Table 2: Parameters used for the STA/LTA detection method**

| Parameter | Value |
|---|---|
| STA window length ($N_s$) | 1 s (1000 samples) |
| LTA window length ($N_L$) | 3 s (3000 samples) |
| Trigger start threshold $\tau_1$ | 2 |
| Trigger end threshold $\tau_2$ | 0.8 |

**3. Results**


Figure 3 shows examples of raw and unprocessed strain rate data measured with the DAS unit in the well at the start of cold-water injection (1st subplot), one hour after start of fluid injection (2nd subplot) and shortly after the end of the 24 hours water injection period (3rd subplot). Each subplot depicts 10 seconds of data with the same data color scaling. A number of features





can be recognized in each of the data examples. At the depth marked with the arrow "A", there is a transition from a noisy
depth interval above to a rather quiet one below. The transition marks the location of the water table in the well. From the
wellhead, the water free-falls down to the water table at about 170 m below surface. In the cased hole section down to the
depth of the transition to the perforated liner, high velocity tube waves (around 1500 m/s) are present which are reflected at
the liner shoe of the 9 5/8" casing at ca. 2810 m MD (arrow "B" in first subplot). Below "B", the cable is located inside the
perforated liner. The tube waves are not further guided in this interval and the noise level is rather low. In the uppermost 100
m of the perforated liner section (2810 – 2900 m MD), a strong signal is present in the 2nd and 3rd subplot (arrow "C"). The
arrow "D" marks another common characteristic feature in the DAS data which was observed over the analyzed cold-water
injection period. This abrupt and localized signal is interpreted as a sudden contraction of the sucker rod.

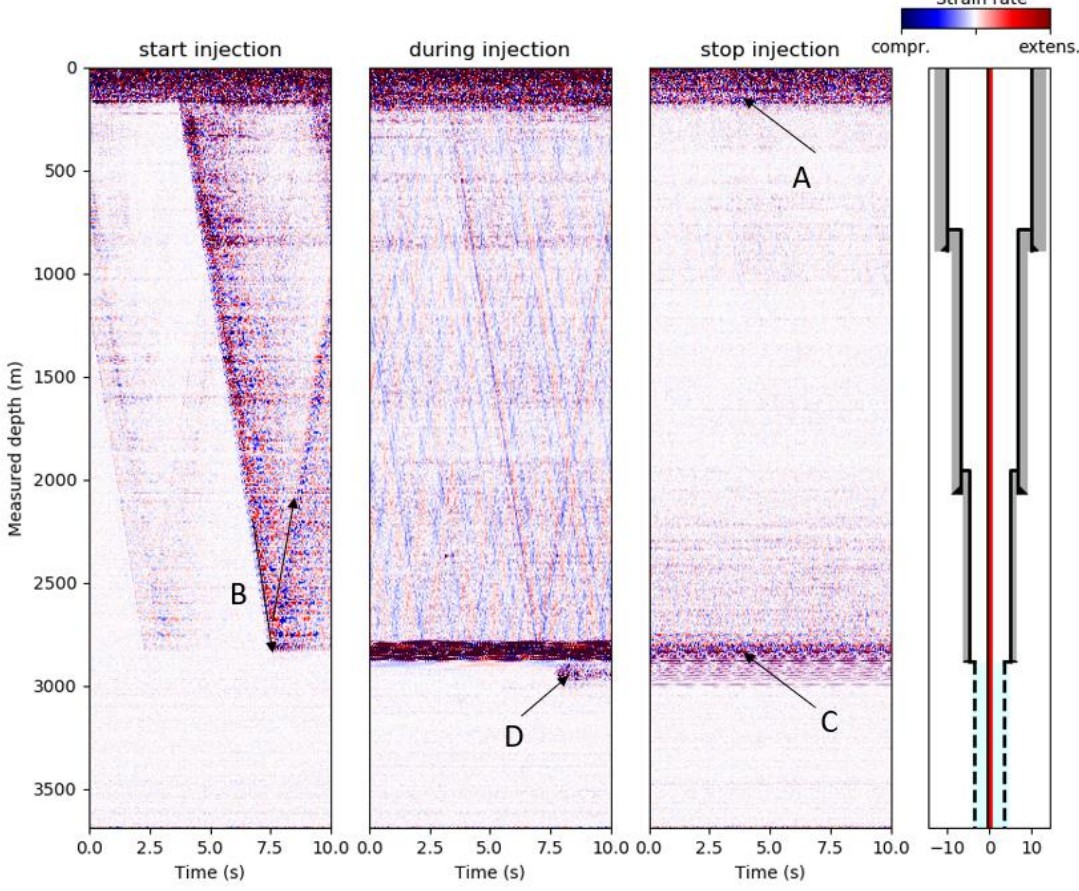

**Figure 3: DAS raw data examples over the scope of the cold-water injection phase for (1) the onset of fluid injection (2) ongoing injection and (3) termination of fluid injection. Blue colours show relative compression and red colours relative expansion. The color ranges are the same for all subplots**





## 3.1 Sucker rod contraction

Figure 4 shows fiber-optic data from DTS and DAS for the first hour of cold-water fluid injection testing. The first subplot
shows three DTS profiles at 00:48, 01:18 and 02:08, which are -8, +22 and +72 minutes relative to the cold-water injection
start. The entire rod from surface to 3100 m experiences cooling. Below the most prominent feed zone of the well at 2830 m
MD, the cooling of the well decreases. This is because most of the injected cold-water flows into the formation (2825 - 2835
m MD) and the fluid column below remained rather undisturbed. A theoretical tensile strain from thermal contraction of the
steel rod (and the fiber-optic cable) $\varepsilon_{\mathrm{DTS}}$ can be derived from the temperature difference between the two profiles for a certain
depth relative to the profile at 00:48. The second subplot compares the 15 m moving average of $\varepsilon_{\mathrm{DTS}}$ calculated after formula
(1) with the local strain ($\varepsilon_{DAS}$) calculated after formula (2) during the same time interval. The third subplot shows the borehole
inclination from the deviation survey. On the fourth subplot, a schematic representation of the casing/liner landing depths is
shown together with the location of the rod.

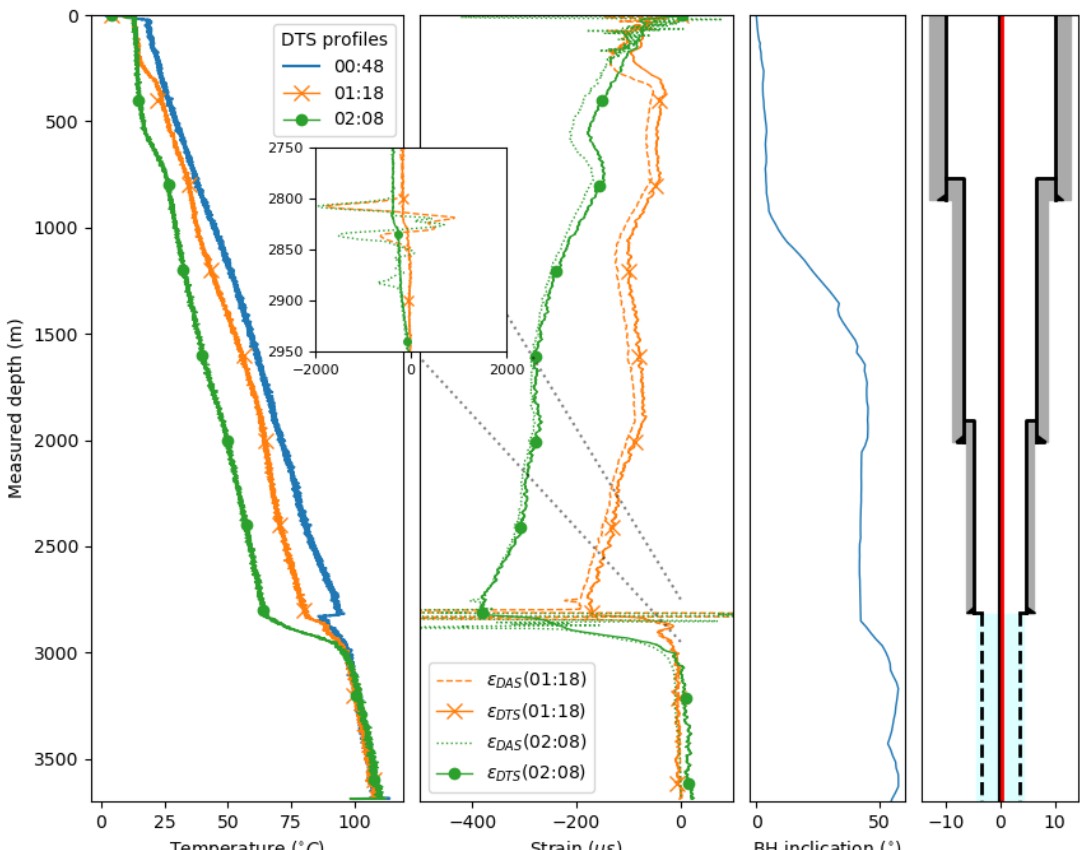

**Figure 4: Downhole monitoring data during the cold-water injection test. 1st subplot: DTS temperature profiles. 2nd subplot:
Comparison of strain profiles ε$_{DTS}$ and ε$_{DAS}$. 3rd subplot: borehole inclination. 4th subplot: wellbore schematic**



In general, a clear match is visible between $\varepsilon_{DTS}$ and $\varepsilon_{DAS}$ for the entire well which means that the strain the steel rod experiences ($\varepsilon_{DAS}$) follows the predicted thermal contraction ($\varepsilon_{DTS}$). However, there are depth intervals where the experienced strain ($\varepsilon_{DAS}$) exceeds and others where it falls short on the predicted strain ($\varepsilon_{DTS}$). Until 2825-2835 m MD where the most prominent injection
interval is located, $\Delta T$ increases with increasing depth. At the injection interval $\Delta T$ rapidly increases. Below this zone, no thermal contraction is expected.

Along the 13 3/8" casing interval (from top liner hanger 13 3/8" at 768 m MD to top liner hanger 9 5/8" 2010 m MD), $\varepsilon_{DTS}$ and $\varepsilon_{DAS}$ are negative and show the same trend thus indicating the expected contraction. In absolute values expected strain $\varepsilon_{DTS}$ exceeds the measured strain $\varepsilon_{DAS}$. Over this depth interval, the well inclination increases from nearly vertical to 45°.

At the transition to the 7" perforated liner at 2810 m MD (top liner hanger packer) a notably different $\varepsilon_{DAS}$ pattern is measured compared to $\varepsilon_{DTS}$ (box plot in Figure 4). In the depth interval 2795-2815 m MD, the expected contraction from $\varepsilon_{DTS}$ at 01:18 yields -170 µε (-380 µε at 02:08), while the estimated contraction from $\varepsilon_{DAS}$ at 01:18 results in -1740 µε (-1950 µε at 02:08) µε between 2805-2810 m MD, which is more than a factor 10 higher (factor 5 at 02:08). In the depth interval 2815-2830 m MD, $\varepsilon_{DAS}$ shows an extension of the rod with a maximum of 900 µε at 01:18 while $\varepsilon_{DTS}$ decreases from -160 µε at 2815 m MD to -
55 µε at 2835 m MD. This is the only locations in which the integrated strain rate from $\varepsilon_{DAS}$ shows extension instead of the predicted contraction. At 2830-2850 m MD, another interval with extraordinary high $\varepsilon_{DAS}$ readings relative to $\varepsilon_{DTS}$ is present. Below 2850 m MD, $\varepsilon_{DAS}$ and $\varepsilon_{DTS}$ again follow the same trend at 01:18. At 02:08, the $\varepsilon_{DAS}$ and $\varepsilon_{DTS}$ show a discrepancy down to 2890 m MD and the same trend below. The gyro data shows a sudden increase in the inclination of the borehole at 2850 m MD. Between 2900-3100 m MD, the temperature difference between the two DTS profiles rapidly decreases (see Figure 4, 1st
and 2nd subplot). Hence, in this lowest depth region of the well, no thermal contraction is expected.

**3.2 Sudden contraction events**

A close-up of raw DAS data is shown for the depth interval 2500-3300 m MD around the transition from cased hole to perforated liner 52 minutes after the start of the cold-water fluid injection (see Figure 5). At this time, the DAS records a transient strain-rate anomaly. Similar events are repeatedly observed in the course of the measurement during the cold-water
injection periods. Using the event shown in Figure 5 as a representative example, we describe common features of these events in the following. Its origin lies at 600 ms and 3000 m MD and is characterized by an abrupt increase of the measured strain rate. The sudden increase of strain rate amplitude propagates both up- and downwards along the well with compressional and tensional sign of amplitude, respectively, where the propagation velocity upwards is approximately 3900 m/s (green Line A in Figure 5). In contrast, the downward propagation velocity is slower and shows irregularities from 650-1260 m/s. Most striking
is the decay of the velocity from 3200 m MD onwards and the eventual stop of propagation slightly above 3300 m MD. In upward direction, this event is halted somewhere in the noisy interval where the reservoir section of the borehole begins. The event is followed by elastic reverberations that decay after approximately half a second.





Further examples of such kind of events are plotted in Figure 6 A, B, C and D. They all have in common, that they originate below 2900 m MD and trigger a contraction above and an extension below. The previously discussed event is characterized

by a smaller precursor 100 ms before the origin of the large event at the same depth. Precursors and successors can also be observed in the examples in Figure 6, yet the events shown here are distinguished by the fact that their upwards propagation extends beyond the noisy reservoir section. All exemplary events except 6A whose downward propagation arrests rather sudden, have in common that the up- and downwards propagation slow down before coming to a halt. Another striking observation in all of the events is that the initial onset propagates slower than the reverberations in the coda.

While the exact shape of the spatial propagation and length varies (length between 20–1600 m), the duration of these events is mostly in the range of 0.5 s with some fading noise/reverberation afterwards. These events typically show a tensional signal at the energy front in the downward direction while the initial energy front upwards is mostly compressional. As the vibrational signal propagates along the rod, a succession of compressional and tensional waves is created which moves with a velocity of about 3900 m/s along the rod (as shown by the green line A in Figure 5). The downward propagation of the first arrival changes

its velocity from the onset of the event towards the end of the vibrational event. In the first 50 ms, it increases in velocity, then it stays constant before it gradually decreases in velocity at around 700 ms below 3200 m MD.

The four black arrows on the left y-axis in Figure 5 indicate the timeseries for which the four spectrograms shown in Figure 7 were calculated with a moving window of 250 ms. The DAS strain-rate timeseries at 3000 and 3200 m MD show the onset of the slip event at 0.5 s with dominant frequencies of the first break between 30 and 75 Hz. The slip only lasts approximately

half a second but reverberations of different duration and different frequencies can be observed in band below 30 Hz depending on the rod segment. For instance at 3000 m MD long lasting reverberations occur at ~10Hz whereas at 3200 m MD they occur at 20 Hz. As can be seen from the spectrogram from the DAS strain-rate recordings at 2700 and 2835 m MD, the slip event does not penetrate into and beyond the feed zone whose characteristic noise at 24 Hz remains undisturbed just as the low frequency pattern of the tube waves above.





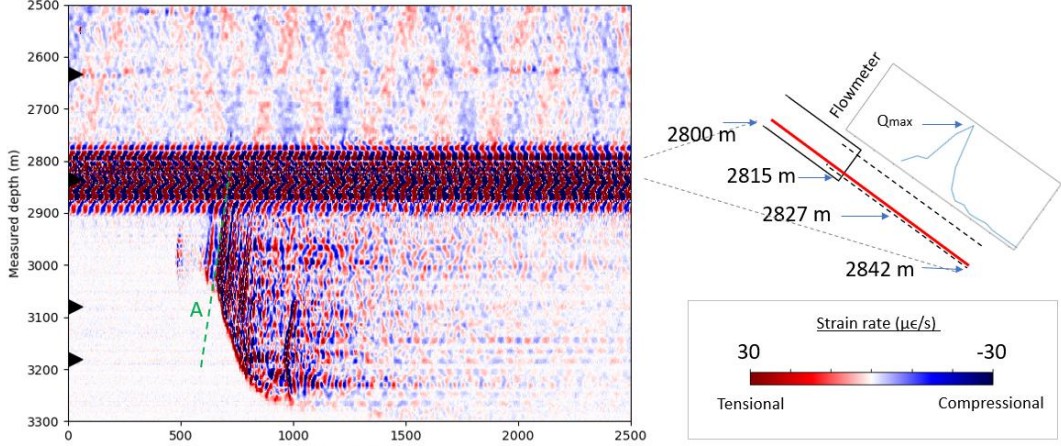


**Figure 5: Sucker rod contraction event displayed by strain rate DAS data (left). The black arrows on the left y-axis mark the depth location of timeseries used for the spectrograms in Figure 7. Line "A" marks the moveout of the signal at a speed of 4000 m/s. The schematic drawing shows the inclination of the borehole with the fiber-optic cable (red) lying inside of the casing (right). The inflow profile from a wireline flowmeter measurement is shown by the blue graph**

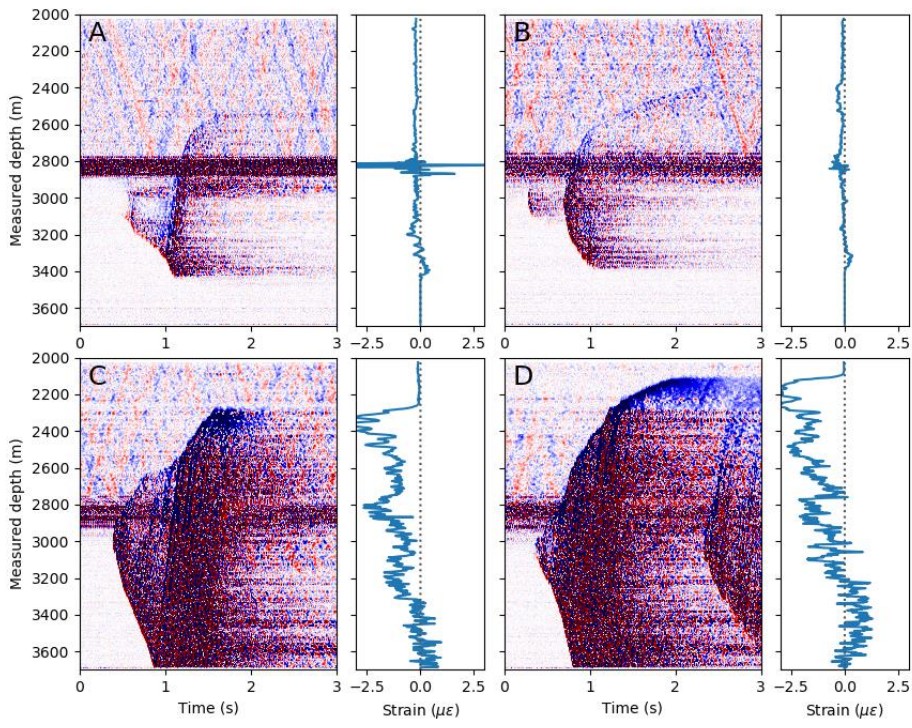


**Figure 6: Four raw DAS data examples of sucker rod events with the integrated strain rate ($\varepsilon_{DAS}$) over a period of 3 seconds. The timing of the events relative to the start of cold-water injection is: + 65 minutes B: + 110 minutes C: + 147 minutes and D: 210 minutes**






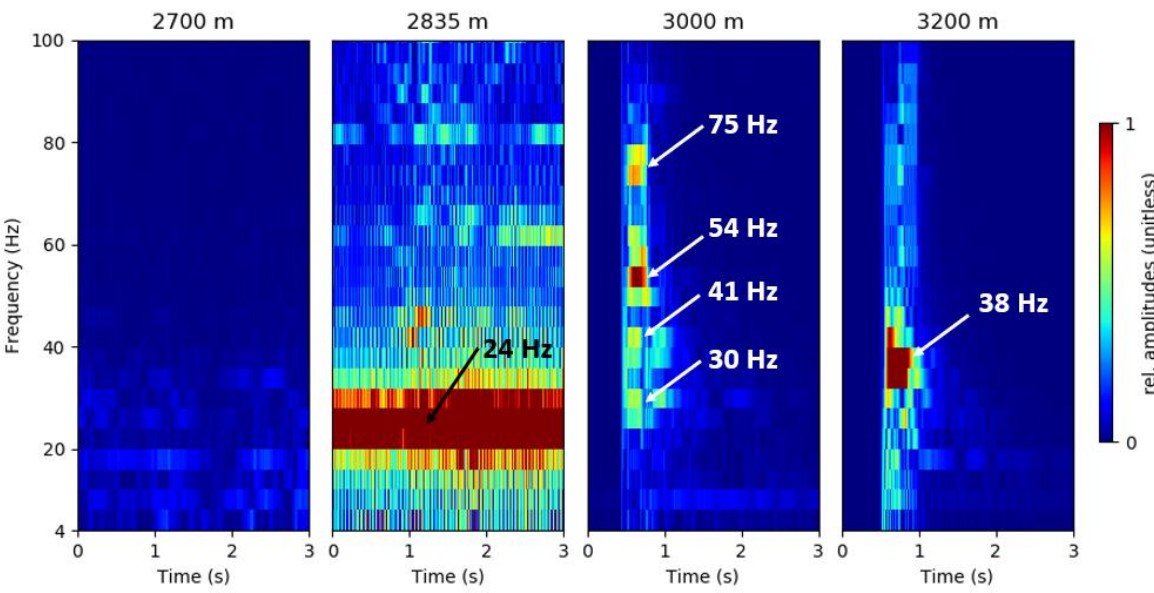

**Figure 7: Spectrograms for a 250ms moving window at different depth along the well during the sudden vibrational event depicted in Figure 5. Red colors indicate high amplitudes, blue colors low amplitudes. The relative amplitudes are displayed by the same color ranges for all subplots.**






### 3.3 Event detection over time

We applied a STA/LTA algorithm to automate the detection of the sudden vibrational events within the first 72 minutes of
cold-water fluid injection. Three attributes are obtained for each event: a) the depth location where the event starts b) the lower
end and c) the upper end of the event according to the STA/LTA algorithm. Figure 8 shows one example of the automated
detection with the STA/LTA trigger. The upper subplot shows an example trace of raw DAS data at a depth of 3120 m MD
(marked by the black arrow in the lower subplot) and the corresponding STA/LTA characteristic function. Beginning and end
of the detection are marked by the green and orange crosses, respectively. The lower subplot shows spatio-temporal DAS data
and the detection of two vibrational events.

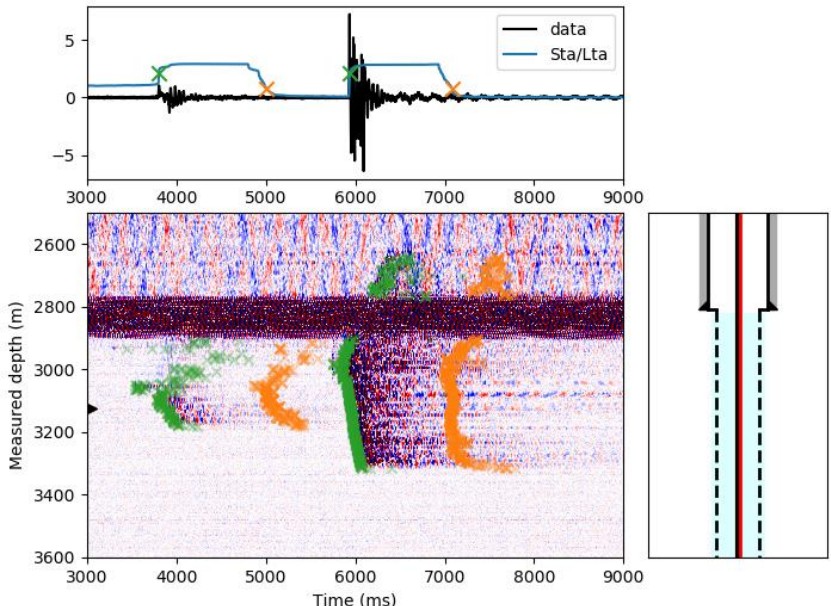

**Figure 8: STA/LTA trigger algorithm applied as an automated detection method for vibrational events. Trigger start and end is marked with green and orange crosses**

All vibrational events which occurred within the first 72 minutes of cold-water fluid injection are plotted in Figure 9. Gray
circles mark the spatio-temporal origin of vibrational events. The corresponding vertical black line indicates the spatial extend
of the respective event. In this representation, events with a spatial extend of less than 20 m are neglected. Such small events
occur between 4-10 times per minute in the depth region from 1250-2750 m MD over the entire investigated 72 minutes after
fluid injection start. Within the first 15 minutes, only a relatively small number of bigger vibrational events occur, i.e. events
which extent over more than 20m. Early events (within the first 5 minutes relative to injection start) appear in the depth region
between 1250-1900 m MD. Except for two large events (4 minutes: 2260-2730 m MD and 6 minutes: 2040-2700 m MD), the
spatial extend of the vibrational events is rather small. One single event was recorded at a depth of 3540-3580 m MD close to





the shoe of the installation. With time, the depth of vibrational events increases to 2900 m MD. From 17 minutes onwards, the occurrence of vibrational events is mostly constrained in the depth region from 2900-3100 m MD. The maximum spatial extend

of large vibrational events increases with time. From 01:18 (+22 minutes after injection start) onwards, most of the events extend into the depth region of 2835-3080 m MD. At 02:08 (+72 minutes to injection start), the spatial extend of the events is 2500-3470 m MD.

With time, the frequency of the occurrence of the events decreases. 4-5 hours after injection start, large events (such as in Figure 6 C and D) appear every 10 – 15 minutes. 8 hours after injection start, large events appear approximately every 25 – 40

minutes.

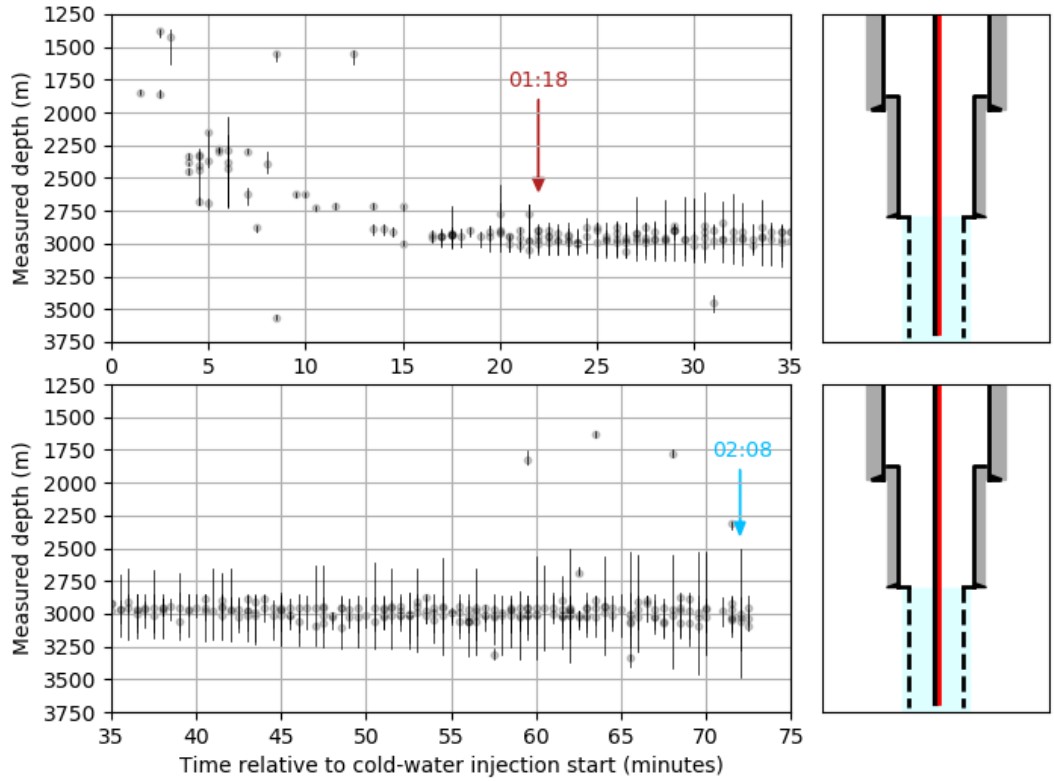

**Figure 9: Gray circles and black vertical lines indicate the spatio-temporal origin and spatial extent of vibrational events in the well, respectively. The shown period comprises the first 72 minutes of cold-water fluid injection.**






### 3.4 Friction force model

The static friction force $F_f$ after formula (3) is compared to the applied force from thermal contraction of the rod $F_{app}$ after
formula (5) which was evaluated for the period from injection start to 01:18 (+22 minutes after start of injection) and to 02:08
(+72 minutes after start of injection) (Figure 10). The gravitational weight force $F_w$ per nylon centralizer is constant for every
contact point of the rod. The force needed to overcome the cumulative static friction $F_f$ is a function of the borehole inclination.
$F_f$ increases from the bottom of the rod installation at 3691 m MD towards 1000 m MD. The bottom end of the sucker rod and
the carrier of the pT gauge at 2755 m MD create an additional static friction force of 0.4 kN and 0.5 kN, respectively. Above
1000 m MD, the well is nearly vertical and only little static friction is expected. The static friction $F_f$ at 1000 m MD yields
26.1 kN. $F_{app}$ at 01:18 is lower than $F_f$ for the entire installation length. Only in the depth interval 2731-2820 m MD, $F_{app}$
approaches a force of 10.5 kN which is close to $F_f$. This indicates that forces are sufficient to initiate relative motion between
sucker rod and casing at that depth. With ongoing cold-water fluid injection, the applied forces $F_{app}$ increase with further
decreasing temperatures. At 02:08, $F_{app}$ surpasses the frictional forces in the depth range from 2150-2912 m MD. $F_f$ and $F_{app}$
intersect at 17.0 kN and 9.3 kN, respectively. At the depth interval from 2732-2820 m MD, the applied force peaks at 22.0 kN
(shown in Figure 10). For all estimates given above, it is assumed that the sucker rod did not move relative to the casing, i.e.,
thermal stresses can build up but will not be released by relative motion.

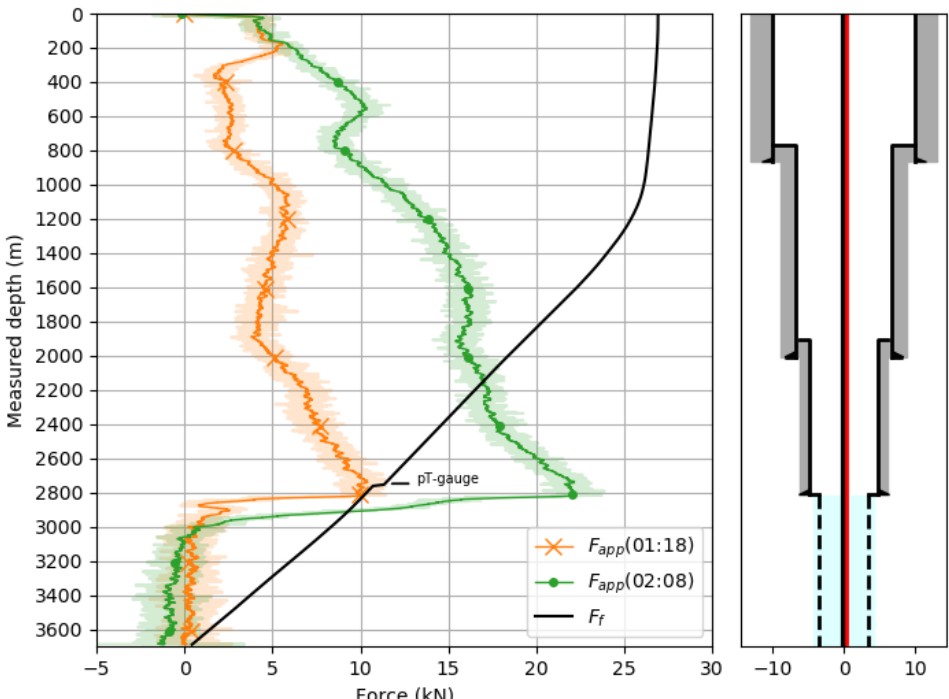

**Figure 10: Comparison of static friction $F_f$ with applied forces $F_{app}$ from thermal contraction of the rod within the first 72 minutes**
**of cold-water fluid injection. The pale colors in $F_{app}$ originate from measured DTS data and the solid lines are constructed by a**
**moving average over 15 m**



## 4. Discussion

With the help of distributed fiber-optic temperature and acoustic data, we monitored a cold-water injection period in a geothermal well at the site Schäftlarnstraße, Munich. The downhole monitoring data allows for an analysis of the deformation

of the 3.6 km long sucker rod/fiber-optic cable construction due to cooling. We observe numerous localized episodes of large strain-rates that nucleate along the inclined stretch of the borehole and propagate both towards greater depth and the surface. Such events induce quickly declining elastic vibrations along the entire extent of the affected interval. The emergence of these vibrational events strongly correlates with the beginning of the fluid injection. In the following, we thus argue that the vibrational events are a result of the substantial temperature changes which the sucker rods with the optic fiber are exposed to.

The contraction of the sucker rods upon cooling induces stress where the sucker rod is held to the borehole wall by frictional forces. On the basis of a simple mechanic model we show that accumulated stresses may eventually exceed the friction giving rise to sudden stress release and the observed strain changes.

### 4.1 Assessment of measuring errors

Our monitoring data analysis is based on a debatable approach of integrating DAS data over longer time periods. To obtain

the $\varepsilon_{DAS}$ profile over the period of 1 hour, a total number of 3.6 million strain rate profiles are integrated (sample rate: 1000 Hz). Such kind of numerical operation has a high risk of creating numerical errors due to e.g. rounding off or value truncation. In addition, the smallest systematic error in the DAS measurement system results in a significant drift over time which would misrepresent the strain profile measured by the sensing fiber. Also, it is well known that for FIMT type of installations, the gel filling allows for a creep and differential movement of fibers with respect to its surroundings which makes strain sensing

unreliable for greater deformations and longer periods (e.g. *Lipus et al., 2018*, *Becker et al., 2020*). However, a creep over many meters or even kilometers is most likely improbable. To strengthen the meaningfulness of our integrated strain profile, we analyzed the $\varepsilon_{DAS}$ for a deeper section of the well, where no (or very little) temperature change was measured by the DTS. In 3500 m depth, we do not observe any strain accumulation after temporal integration of strain rate data over a period of 60 minutes. This indicates that the measured strain rate has no significant drift during the time of interest. For measurements with

higher amplitudes such as within the depth interval 2800-2900 m, non-linear effects influencing the temporal integration of the data cannot be excluded.

### 4.2 DAS data integration

We integrated DAS data in time over 72 minutes to assess the absolute contraction of the rod construction prior to the cold-

water injection start (see Figure 4). For the well interval from water table to the transition to the perforated liner, the results show a good match to the contraction that was theoretically assumed from the cooling of the well. However, from 2800-2900 m MD, we obtain much higher deformation from the DAS data than what we expected. We cannot give an unambiguous



explanation for that but see two likely reasons for that observation. Firstly, the DAS integration process might result in a drift when integrating high amplitude DAS data. Especially from 2800-2900 m MD, constantly high energy is recorded by the

system. The second explanation could be that the integrated DAS data measured a true deformation of the construction. In the depth region around 2800 m, the annular space of the borehole is rather irregular (transition to 7 " liner interval, localized increase in the borehole inclination (see Figure 4, 3$^{rd}$ panel)). The repeating sudden sucker rod events might lead to an uneven distribution of the thermal stresses along the rod. Interestingly, the most prominent feed zone of the well coincides with the one single DAS interval which shows an extensional signal.

The sudden slip events presented in this study show some similarity to the "slip events" which were previously observed in FIMT-type fiber-optic installation in a geothermal well (*Miller et al., 2018*). In the reported DAS campaign, a fiber-optic cable was installed in a geothermal well and it is argued that repeated thermal cycles led to a loss of frictional coupling between fiber-optic cable and the borehole wall. *Miller et al., 2018* reported that a sudden loss triggered a movement of the cable with a first arrival speed of 4600 m/s (we measured a first arrival speed of 4000 m/s). The integrated strain of the reported event

shows a balance towards absolute contraction which we also observe in our events. Another similarity is given by the frequency content of these events. They recorded a dominant frequency of 45 Hz with some harmonics in both directions which we also observed in our data (see Figure 7 at 3000 m MD).

**4.3 Stick-slip rod behaviour**


We calculated the static friction force $F_f$ along the rod construction by a cumulative sum of the friction of each nylon centralizer with the borehole inner wall. Independently of that, we computed the applied force $F_{app}$ on the rod construction by thermal contraction using the DTS monitoring data. By comparing both curves, we can distinguish depth regions where the rod remains immobile ($F_f > F_{app}$) and depth regions where the applied forces overcome the static friction force ($F_f < F_{app}$). The temperature

difference in the course of the investigated time period is particularly high over the 9 5/8" liner interval (depth region from 2485-2890 mMD) which in consequence also means that $F_{app}$ is high. According to our model calculation, the contraction forces surpass the frictional forces at 2800 m MD around 01:18 (22 minutes after injection start). This result implies that after this time, the construction can contract in this depth interval. In other words, the thermal stresses on the rod construction in this depth region are high enough that the rod starts to move and to contract.

With ongoing cold-water injection and further cooling of the well, the applied forces $F_{app}$ increase. This leads to a continuous growing of the depth interval where $F_{app}$ surpasses the static friction $F_f$ of the rod. The STA/LTA detections match the predictions of the friction fore model. After a rather quiet initial phase of low energetic events (before 17 minutes in Figure 9) which could be caused by the relaxation of previously accumulated stress anomalies along the sucker rods, repeated vibrational events start to concentrate in the region 2800-3100 m MD. As the region with $F_f < F_{app}$ increases, the length of the vibrational

events increases. From our friction force model, we would expect vibrational events (more specifically: the contraction part of the movement) at 02:08 in the depth region 2150-2910 m MD. However, the observed events extend from 2500-3500 m MD.



Regarding the upper limit, we can see in Figure 10 that there is a significant change in slope for $F_{app}$ at 02:08 at 2500 m MD. The friction force model is based on numerous assumptions (i.e. static friction coefficient nylon-steel, Young's modulus for stainless steel, neglecting fiber-optic cable, stress-free initial conditions) which might not accurately depict the downhole
conditions. This could mean, that either the calculated applied force $F_{app}$ is too high and/or the static friction force $F_f$ is too low. With respect to the lower limit of the vibrational events, we predict the contraction part ($F_{app}(02:08)$ Figure 10) of the vibrational events down to a depth of 2912 m MD. However, we record vibrational events down to a depth of 3480 m MD. This discrepancy can partly be explained by the fact that the model prediction only shows the contraction part of the vibrational event. As seen in the cumulative strain $\varepsilon_{DAS}$ (see Figure 6 event A and B), the lowest part of
a vibrational event yields extension. The most likely reason is that the contraction above results in a pulling of the rod from a lower lying region to compensate for the missing rod length. Therefore, the events can be traced down to a greater depth than predicted.

## 5. Conclusion

The field test at the geothermal site Schäftlarnstraße demonstrates that simultaneous recording of DTS and DAS data can be
used for a detailed analysis of the deformation of a sucker rod type of fiber-optic cable installation in a 3.6 km deep well. By comparing the theoretical contraction of the rod structure from DTS with an estimated contraction from DAS, we can distinguish depth intervals with higher and lower thermal stresses in the material. We introduce a friction force model which accurately predicts the onset and extent of sucker rod events releasing accumulated thermal stress. This is an important finding for DAS monitoring in geothermal settings because it shows that localized high-energetic vibrational events must not
necessarily be related to microseismic events occurring in the rock formation but can originate in the subsurface construction and the way how the FO monitoring equipment is installed in the well. Moreover, the friction force model is useful to predict the data quality for DAS measurement campaigns for deep sucker-rod types of FO installations. Especially for the recordings of weak acoustic signals that are e.g. induced by fluid movement in the annulus, it is essential to know the potential sources of errors and artifacts in the data. During operations which introduce a temporal temperature gradient, thermo-mechanical
response of freely hanging steel parts in the borehole may introduce stick-slip events that must be distinguished from any other relevant seismogenic source. Potentially, the vibrational energy from the sucker rod events can also be used to study the formation velocity in the near-field around the borehole. Furthermore, the large-scale contraction along certain sucker rod and fiber intervals must be considered with respect to the location of the distributed sensor nodes. Our description also serves a

starting point for a more detailed dynamic description of the observed processes. This can be of use to predict onset and depth
intervals of such sucker rod events and to contain their destructive potential in case of too quick cooling of the construction.

**Code and data availability**

Python scripts and data are available upon request to the corresponding author.

**Author contributions**

TR and KZo conceptualized, planned and coordinated the monitoring campaign. MPL, FS, CW, TR and DP conducted the
field measurement. MPL performed the DAS data processing. All authors contributed in the interpretation of the results. MPL
prepared the first draft of the manuscript with the contribution from all authors.

**Competing interests**

The authors declare that they have no conflict of interest.

**Special issue statement**

This article is part of the special issue "Fibre-optic sensing in Earth sciences". It is not associated with a conference.

**Financial support**

The fiber optic cable was installed in the framework of the Geothermal Alliance Bavaria project, funded by the Bavarian
Ministry of Economic Affairs, Energy and Technology. A part of this work was financed by the GeConnect project
(Geothermal Casing Connections for Axial Stress Mitigation), coordinated by ÍSOR, which is funded through the ERANET
cofund GEOTHERMICA (Project no. 731117), from the European Commission, Technology Development Fund (Iceland),
Bundesministerium für Wirtschaft und Energie aufgrund eines Beschlusses des Deutschen Bundestages (Germany) and
Ministerie van Economische Zaken (The Netherlands).

**Acknowledgements**

This work would not have been possible without the continuous support from our partners involved in the project. The authors
are thankful to Stadtwerke München, owner and operator of the geothermal site Schäftlarnstraße, for providing access to the
well site, their premises and well data. The authors would also like to thank the drilling contractor Daldrup for accessing the
site during the drilling and well completion operation. Moreover, we would like to give credit to our colleagues at Erdwerk



GmbH and Baker Hughes for the close collaboration and fruitful discussions. From GFZ, the authors are thankful to Christian Cunow and Tobias Raab who supported the field work and acquisition of fiber-optic data.




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
