# Peer review of "Dynamic motion monitoring of a 3.6 km long steel rod in a borehole during cold-water injection with"

_Solid Earth, 2021_

## Author Comment (AC1)

**Author comments to the reviewers:**

Potsdam (Germany), the 20th of September 2021

Dear Mr. Ryan Schultz,

Thank you for your time and effort in reviewing our publication. Your input and suggestions are valuable to us. Below, you find our replies (in grey) to your comments (in black).

Best regards,

Martin Lipus

**Reviewer 1:**

Review of Solid Earth article MS# 2021-63,

The manuscript of Lipus et al., "Dynamic motion monitoring of a 3.6 km long steel rod in a borehole during cold-water injection with distributed fiber-optic sensing" is an article concerning the observation of thermally induced stick-slip events. Cold water pumped into a well causes the rod inserted into the well to contract, with shaking from the stick-slip events recorded on DAS. The results are compared against DAS and DTS data to build a picture of where and why these events are occurring in the well. I think that the results of the paper could be interesting the readership of Solid Earth.

For this paper, I have only a few small critiques that should be addressed before acceptance with Solid Earth. In general, my comments revolve around better explaining some of the arguments the authors are trying to make. A more detailed list of my thoughts follows below:

In the conclusion, the acronym FO is used for the first time. I'm assuming it means fibre optic. I'd recommend removing it, as it's only ever listed here.

Acknowledged. Thank you.

From the results of this paper, it should be possible to get a rough estimate of what the coefficient of friction is between the well and the rod. Would be interesting to get a back-of-the-envelope sense of what that value is.

The presence of the first events coincides with the temperature, the theoretical model predicts its occurrence. Therefore, the literature values assumed for the static friction between sucker rod and steel liner are assumed to approximate the real values. A note was added to the discussion:

Was:

"In other words, the thermal stresses on the rod construction in this depth region are high enough that the rod starts to move and to contract."

Now reads:

"In other words, the thermal stresses on the rod construction in this depth region are high enough that the rod starts to move and to contract. Hence, the literature values assumed for the static friction between sucker rod and steel liner are assumed to approximate the real values."

In Section 4.1 the authors talk a bit about errors that apply in the measured and expected strains. I'm curious as to what sorts of errors could be introduced from the DAS data based on the response spectrum of the fibre optic cable. Are we potentially attenuating frequencies that could contribute significantly to the measured strain?

In the upper part of the well, the calculated strain derived from the DAS data shows variations related to the temperature change measured from the optical fiber. The change in strain is visible as an offset from zero in the strain rate data. In the reservoir section of the well, the strain development coincides with the stick-slip events. In the deepest part of the well, no temperature change and no strain change were observed; no offset from zero was measured in the strain rate data due to the temperature change. This led to the following conclusions:

Assuming that any vibration would lead to a variation of strain rate data around zero (+offset due to temperature changes), possible attenuation phenomena for specific frequency bands related to the installation design should not contribute to the overall strain change. Only if very low frequencies (<<1Hz, similar or lower than the temperature changes) were attenuated, this could have an effect. We find it unlikely that such low frequencies can occur.

To verify that there are no drift phenomena of the measurement system influencing the result, we analyzed the lower part of the cable where no temperature change was observed. Here, we also did not find any strain change.

As the system acquired data at 10kHz and downsampled it to 1kHz, higher frequencies might be visible as a beat frequency (that might be interpreted as a very low frequency signal). We do not expect a significant contribution at such high frequency.

The following page lists minor corrections and typos to be fixed.

Thanks,

-Ryan

Near Lines 372 & 377 & elsewhere: "extend" should be "extent"

Acknowledged. Thanks!

Figure 9: Why do events only seem to occur at the top of the casing liner after ~16 minutes?

We can name five effects that might contribute to the question why the events occur in particular at the top of the casing liner. Firstly, along the entire well path, the cooling from fluid injection is highest in this depth interval close to the top of the liner, where the most prominent geological feed zone lies. Consequently, the calculated force from thermal contraction is highest here (orange graph in Figure 10). Below that depth, the thermal contraction steeply decreases. The cummulative static friction of the rod construction is lowest at the bottom of the well and increases towards the surface (black graph in Figure 10). Both orange and black graph happen to "meet" around ~16 minutes after injection start close the top of the liner zone. Therefore, many events occur in this depth - and at that particular time. Secondly, the static friction force of the rod gradually increases towards surface, meaning that it is less and less likely for any relative motion to occur at shallower depth. Thirdly, we see that the gyro recording shows a sudden increase in the inclination of the borehole at 2850 m MD (see third panel in figure 4) and consequently an increase in F_N due to the higher angle. Effect No. 4 is that there is a change in diameter at the liner hanger. Effect no. 5: Above the top if the casing liner, fluid moves along the rod resulting in motion of the rod and hence sliding friction. Below, there is only static friction left that must be overcome first before shallower sections of the rod can move.

Section 2 rework:

**2. Data Analysis**

[revised manuscript text omitted]

---

## Author Comment (AC2)

**Author comments to the reviewers:**

Potsdam (Germany), the 20th of September 2021

Dear colleague,

Thank you for your time and effort in reviewing our publication. Your input and suggestions are valuable to us. Below, you find our replies (in grey) to your comments (in black).

Best regards,

Martin Lipus

**Reviewer 2:**

This manuscript reports interesting observations in a borehole with DTS and DAS, which includes sucker rod contraction and sudden contraction events. The sudden contraction events on the DAS records during the first 72 minutes of cold-water fluid injection are reported. A friction force model was proposed to explain the mechanism of vibration events instead of the microseismic event. General speaking, it provides very useful information. The following is my comments.

Section 2.

Eq 1 should be in "2.1 derivation of strain from Distributed Temperature Sensing" rather than "2.3 Deformation balance …".

Agreed. The explanation of the method and the related equation are placed in succession to improve readability. We added the equation closer to the DTS description.

"2.2 Direct measurement of strain via DAS". iDAS measures the strain rate instead of strain. Was any high pass filtered applied to the raw data?

No high pass filtering was applied to the raw data. We have added a sentence to clarify. At the end of 2.2 Direct measurement of strain via DAS now reads:

"No additional filtering was applied in post-processing (no high pass and no low pass filtering)."

"2.3 Deformation balance from DTS and DAS measurements". This section includes how to compute strain from DTS and obtain strain from strain rate records, which is not strongly related to the "balance". It may be better to merge with section 2.1 & 2.2.

To improve the readability of the method section, we restructured the method part. The subchapter 2.3 Deformation balance from DTS and DAS Measurements" was removed from the manuscript. Instead, it was merged with section 2.1 and section 2.2.

Eq. 5, the applied force Fapp is used in this study instead of the stress.

Thank you for pointing this out. You are correct, it makes more sense to introduce the applied force Fapp instead of the stress. The text is changed accordingly.

"2.4 Event detection and picking" looks not related to the other sections.

We have changed the title of the subchapter to anticipate the significance of this additional tool for the analysis of the fiber-optic recording. It now reads:

"Stick-slip event detection and picking"

Section 3.

Line 290-295, the difference between strain_DTS and strain_DAS looks relative to the inclination angle. It may be worth to make a figure showing this difference and inclination angle. Adding some discussions about this phenomenon is also useful. Another interesting observation is that difference at 01:18 is larger than the one at 02:08, especially between ~700m and 2800m.

That is an interesting observation. Thank you for pointing this out. Below, we computed the difference for strain_DTS and strain_DAS of Figure 4 (see Figure 4_review). The most prominent apparent relation between strain and borehole inclination is located at the top of the liner, where the inclination strongly increases in a downward direction from an angle of 42° at 2852 m MD to a value above 50° below 2950 m. This is already discussed in the manuscript. To make this point clearer to the reader, we have added the borehole inclination to the close-up plot in figure 4.

For the remaining differences in the fiber optic strain readings along the well path, we do not see a strong correlation to the borehole inclination.

However, we might make a statement about the strain differences of Figure 4_review with respect to time. The strain difference above 2800 m MD is higher at 01:18, because the cable is stretched to a maximum before the sucker rod events occur. Within the next 50 minutes (until 02:08), the sucker rod events lead to a relaxation of the cable and therefore the strain difference reduces.

For the depth interval below 3100 m, please see the following comment.

[Figure]

*Figure 4_review: Strain differences between DTS and DAS*

On the 2nd subplot of the Figure 4, the differences between strain_DTS and strain_DAS below the 3100m MD are quite different at 01:18 and 02:08. At 02:08, the strain_DTS is positive while the strain_DAS is close to zero. Such difference is not observed on the data at 01:48. Any clue?

Thank you for spotting this. We overlooked this phenomenon in this depth interval. We carefully checked the DTS data and found that for the deeper part of the well (deeper than 3100 m MD), a constant offset in the DTS profiles by about one degree Celsius in subsequent measurements is present. Such offset is not observed in the shallower part of the well. Also, no anomaly is observed in the P/T gauge data from 2750 m MD and no anomaly is observed in the DAS data. We speculate that the temperature anomaly is related to the processing of the DTS data. DTS temperature was measured in a double-ended configuration. A temperature profile is created by overlaying the DTS signal from both directions which are measured consecutively for both fiber branches. Close to the folding location (at the bottom of the well), an asymmetry in the temperature reading was observed between both fiber branches, which does not seem to be caused by any fluid motion. Averaging this difference between both branches led to a temperature offset. This offset was only visible if strong temperature changes were observed in the upper part of the well.

Was:

"Between 2900-3100 m MD, the temperature difference between the two DTS profiles rapidly decreases (see Figure 4, 1st and 2nd subplot). Hence, in this lowest depth region of the well, no thermal contraction is expected."

Now reads:

"Between 2900-3100 m MD, the temperature difference between the two DTS profiles rapidly decreases (see Figure 4, 1st and 2nd subplot). At 02:08, the DTS profile shows slightly increased temperatures (+1 °C) with a constant offset from 3100 m to the end of the cable compared to the DTS profile at 01:18. This leads to a constant offset of a positive expected strain $\varepsilon_{DTS}$. The measured strain $\varepsilon_{DAS}$ shows no offset in this depth interval."

A paragraph was added to the discussion:

"The constant temperature offset by +1 °C in the DTS profiles from 02:08 (relative to 01:18) in the depth interval from 3100 m MD to the end of the cable is unlikely to be caused by any fluid movement. While DTS temperature measurements did show a variation, no additional offset was recorded from the measured strain εDAS. This could mean that the rod builds up thermal extensional stresses without actual movement taking place (εDTS > 0 εDAS = 0). However, we speculate that the temperature anomaly is related to the processing of the DTS. DTS temperature was measured in a double-ended configuration. A temperature profile is created by overlaying the DTS signal from both directions which are measured consecutively for both fiber branches. Close to the folding location (at the bottom of the well), an asymmetry in the temperature reading was observed between both fiber branches, which does not seem to be caused by any fluid motion. Averaging this difference between both branches led to a temperature offset. This offset was only visible if strong temperature changes were observed."

**Since both section 3.2 and 3.3 reported sudden contraction events, it is possible to merge together.**

That is true. We have merged section 3.2 and section 3.3. Because these chapter are already quite extensive, we have added two sub headers in the new section 3.2 "Event description" and "Event detection over time".

**Line 320. It is not easy to see precursors and successors on the Figure 6. Add marks on the Figure 6?**

The statement that precursors and successors are also present in the events in Figure 6 is somewhat misleading and inaccurate. It is only clearly visible in the subplot Figure 6 B. The text was changed accordingly to clarify.

Now reads:

"Precursors and successors can also be observed in the examples in Figure 6 (in particular in Figure 6 B), yet the events shown here are distinguished by the fact that their upwards propagation extends beyond the noisy reservoir section."

As shown in Figure 8, some STA/LTA detections are outliers. How to determinate the origin time of each event and the origin depth? Another interesting parameter is the strength of event. Does the stronger event have stronger spatial extend?

Looking at all sucker rod events as a whole within the first hour of fluid injection, their appearance and shape is highly variable. As accurate automated picking was out of scope for this study, the picking of the depth and shape was done manually by displaying each 30 seconds of raw DAS data recording overlaid by the trigger start and end marker as shown in Figure 8. The onset time/depth location was picked at the based on the moveout of the signal towards top and bottom. The upper and lower boundaries are picked when the Sta/Lta stops to trigger.

Section 2 rework:

**2. Data Analysis**

[revised manuscript text omitted]